# A Framework for Sample and Objective Forgetting: Pull-to-Outlier & Contrastive Objective-level (POCO) Unlearning

## Abstract

Current Machine Unlearning (MU) methods require full retraining or extensive fine-tuning, lack formal removal criteria, and focus only on sample-level forgetting, limiting their practicality. We address these gaps with two lightweight, projection-only techniques operating above frozen feature extractors. Pull-to-Outlier Unlearning (POU) offers a transparent, unsupervised geometric removal method by displacing embeddings of unwanted samples or entire classes into synthetic outlier regions, while preserving downstream performance and distilling knowledge of the remaining data. To the best of our knowledge, Contrastive Objective-level Unlearning (COU) is the first method to remove learned objectives. It perturbs projection weights to eliminate a target task's influence. Then it realigns the original data manifold, which can provide the possibility for managing agentic learning behaviors. We validate POU on CIFAR10, CIFAR100, and Caltech-256 with ResNet-based backbones, showing efficient instance and class forgetting with minimal impact on retained accuracy. COU is tested on DINO and CLIP feature representations, demonstrating effective objective-level erasure while preserving all non-target tasks.

## 1 Introduction

Machine unlearning (MU) refers to a family of techniques that enable the selective removal of specific training data or learned behaviors from a trained machine learning model, without the need to retrain the model from scratch. The motivation for MU arises from the growing legal, ethical, and operational requirements in real-world deployments. For instance, the European Union's General Data Protection Regulation (GDPR), particularly Article 17—the "right to erasure"—mandates that individuals must be able to request the removal of their personal data from any system that has used it (European Parliament and Council, 2016). Similarly, the California Consumer Privacy Act (CCPA) enforces users' rights to request deletion of their information, creating a pressing need for compliant and practical unlearning mechanisms in AI systems (California State Legislature, 2018).

Beyond legal mandates, MU is increasingly essential in sensitive deployment contexts. In cybersecurity, models must revoke the influence of adversarially poisoned or malicious data that could compromise the safety of predictions. However, studies reveal that traditional unlearning techniques often fail to fully mitigate such attacks, underscoring the need for more resilient strategies (Pawelczyk et al., 2024). In healthcare analytics, institutions must ensure that withdrawn patient records no longer influence clinical predictions, especially when consent is revoked after the data have already been used (Sakib & Xie, 2024). Federated learning scenarios present additional challenges: individual clients may request removal of their contributions after global aggregation, necessitating efficient and non-disruptive unlearning methods (Wu et al., 2023). These examples collectively highlight that MU is not only a theoretical safeguard but a practical necessity for deploying AI in dynamic, privacy-sensitive environments.

Despite this progress, existing MU methods share four key limitations:

Figure 1: Pipeline of the proposed Pull-to-Outlier Unlearning (POU) method. The process begins with feature extraction using a frozen encoder, followed by prototype selection via clustering. Unlearned samples are then mapped to synthetic outlier targets using expert-specific projection heads, while a frozen loss preserves the structure of retained samples. Final outputs are aggregated for evaluation.

1. They primarily focus on *sample-level* or *class-level* removal, offering no systematic mechanism to forget an entire learned objective or capability embedded within a model's representation.

2. They do not provide a transparent, per-example verification mechanism to determine whether unlearning has been successfully achieved. As a result, many methods rely on statistical heuristics or indirect metrics to assess success.

3. Existing methods typically require full or partial fine-tuning of the backbone network, whereas our projection-only approach avoids backbone updates entirely, offering significant computational advantages.

4. Only a limited number of approaches, such as Label-Agnostic Forgetting (LAF), support fully *unsupervised* forgetting, where the method can operate without class labels or supervision during the unlearning process (Shen et al., 2024).

To address all four of these challenges, we propose two lightweight unlearning methods that operate entirely above frozen feature extractors and, therefore, avoid any modification to the pretrained backbone:

- **Pull-to-Outlier Unlearning (POU)** is designed for scalable, unsupervised forgetting of specific samples or entire classes. It works by pushing the embeddings of targeted samples outside the known data manifold, making them geometrically incompatible with downstream use. POU leverages a mixture-of-experts (MoE) framework: retained embeddings are first clustered, and each cluster is anchored by a prototype. For every sample to be unlearned, a small residual MLP "expert" is assigned and trained to map it to a synthetic outlier vector located beyond the global min–max bounds. To preserve the rest of the representation space, a frozen-loss term keeps all other embeddings near their original positions. This modular design enables scalable deletion across large sets without updating the backbone, while also offering a transparent test, based on geometric displacement, to verify that the unlearning has succeeded.

- **Contrastive Objective-level Unlearning (COU)** addresses a higher-level goal: the removal of an entire learned capability, such as a class or behavior, rather than individual samples. It begins with a projection head trained using supervised contrastive learning to encode task-specific semantics. COU then selectively perturbs this projection to collapse the representation of the target task, effectively erasing its contribution. A lightweight fine-tuning step on retained data restores class separability for all remaining tasks. Unlike prior instance-based or class-based methods, COU performs this forgetting entirely through projection weights, enabling the first objective-level unlearning without modifying the feature extractor. This makes it particularly suitable for scenarios requiring selective capability removal, such as the moderation of emergent behaviors.

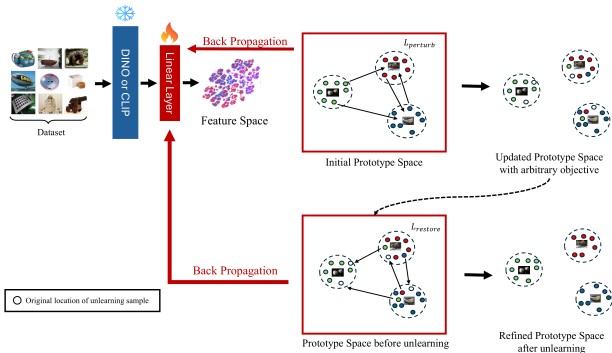

Figure 2: Pipeline of the proposed Contrastive Objective-level Unlearning (COU) method. A projection head is first trained on top of frozen backbone features using supervised contrastive loss. After a new objective is externally introduced by modifying class associations, COU unlearns this objective by pulling affected embeddings back to their original positions, restoring the original structure of the representation space.

## 2 RELATED WORK

### 2.1 MACHINE UNLEARNING

Early efforts in machine unlearning focused on exact removal for simple models. One such foundational approach proposed an algorithm for summation-based learners that could analytically revoke the influence of specific training points, but this technique did not extend to modern deep architectures (Cao & Yang, 2015). While full retraining remains the most accurate method to ensure data removal, it is computationally impractical for large-scale models and datasets. To mitigate this, partition-based approaches like SISA (Sharded, Isolated, Sliced Aggregation) training were introduced, which divide the training data into multiple disjoint subsets. When a removal request is issued, only a subset of these partitions require retraining, significantly reducing computational burden while maintaining fidelity (Bourtoule et al., 2019). Another line of work approximates the unlearning process by reversing stochastic gradient descent updates. UnrollSGD simulates unlearning by backtracking the most recent optimization steps, enabling partial reversibility of training with bounded approximation error (Thudi et al., 2021).

In contrast to these strategies, some methods attempt to remove information by adjusting model gradients directly. One such method, commonly referred to as NegGrad, applies updates in the negative gradient direction of the samples to forget. While effective in some cases, this approach can cause undesirable shifts in the representation of non-target samples, especially when fine-tuning deep backbones. More recently, unsupervised strategies have emerged. Label-Agnostic Forgetting (LAF) addresses the removal of entire classes without relying on class labels. It uses a variational distribution matching mechanism to suppress the influence of target distributions, enabling scalable and label-free forgetting in deep neural networks (Shen et al., 2024).

Recent works have explored contrastive learning objectives to improve machine unlearning. For instance, (kyu Lee et al., 2024) proposes to push embeddings of the data to be forgotten away from their original classes and toward alternative representations, modifying InfoNCE-style objectives. (Wang & Chen, 2024) introduces gradient-based constraints to diminish the influence of selected data points in contrastive and supervised learning settings, requiring only a few fine-tuning steps. (Wang et al., 2024) focuses on the auditing aspect by calibrating alignment metrics within contrastive models to verify successful removal without full retraining. While these methods offer improvements at the sample level, they still require backbone updates and do not support objective-level unlearning.

### 2.2 PROTOTYPICAL LEARNING

Prototype-based methods represent each class with a centroid in the embedding space, offering interpretability (Angelov & Soares, 2019; Angelov et al., 2025; Chen et al., 2018; Rymarczyk et al.,

2021) and robustness, especially in few-shot learning (Snell et al., 2017). Prototypical Networks classify by measuring distances to these centroids. Clustering-based approaches like DeepCluster alternate between k-means (Lloyd, 1982) and classification via pseudo-labels (Caron et al., 2018), while SwAV uses online cluster assignments to enable large-scale self-supervised learning without labels (Caron et al., 2020).

These prototype-based strategies also support open-set recognition and anomaly detection by modeling in-distribution regions. Our POU method leverages this by (i) using fixed prototypes as anchors for generating synthetic outliers, and (ii) preserving retained samples via a frozen-loss objective—thus maintaining the data manifold while selectively removing targets.

## 2.3 OUTLIER DETECTION

Outlier detection identifies data points that deviate from the underlying distribution. Classical methods include z-scores, Tukey's fences, LOF for density comparison (Breunig et al., 2000), and DB-SCAN for detecting sparse regions (Ester et al., 1996). With deep learning (Pang et al., 2020b), approaches, such as one-class SVMs (Hearst et al., 1998; Schölkopf et al., 1999), Autoencoders (Rumelhart et al., 1986), and GANs (Goodfellow et al., 2014) have been widely adopted for high-dimensional settings. Surveys by Chandola et al. and Pang et al. highlight this shift toward learned representations (Chandola et al., 2009; Pang et al., 2020a).

POU builds on these ideas by generating synthetic outlier targets beyond global feature bounds and using $\sigma$-band checks around cluster prototypes to transparently verify that unlearned embeddings lie outside valid class regions.

## 3 METHODOLOGY

### 3.1 PRETRAINING AND FEATURE EXTRACTION

For the POU, we begin by pretraining a ResNet-18 backbone (He et al., 2015) followed by a three-layer residual MLP classifier using standard cross-entropy loss on the full training set. The residual MLP acts as a lightweight head and forms the projection module for POU. During unlearning, we continue updating this same MLP head while keeping the ResNet backbone frozen. This setup ensures that the learned embedding space remains stable, and the unlearning process is confined to the projection layer alone. Since POU is designed to operate in a label-free manner, this pretraining step serves to construct a semantically meaningful feature space that remains decoupled from the downstream forgetting mechanism.

For COU, we use pretrained feature extractors from DINOv2 (Caron et al., 2021; Oquab et al., 2023) and CLIP (Radford et al., 2021) ViT-L/14 to obtain high-level representations of input images. On top of these frozen vision transformer backbones, we train a linear projection head using supervised contrastive loss. This loss encourages intra-class compactness and inter-class separation in the projected embedding space, establishing the structure needed for selective objective-level removal. The linear head is later updated to remove the influence of a specific objective, while the backbone remains unchanged.

### 3.2 INITIAL PROTOTYPE SPACE

Given a set of feature embeddings $\{z_i\}_{i=1}^N \subset \mathbb{R}^d$ extracted from a frozen encoder and subsequent projection layer, we apply $K$-means clustering to partition them into $C$ clusters. Let $\{\mu_c\}_{c=1}^C$ denote the resulting cluster centers, and let $c(i) \in \{1, \ldots, C\}$ be the cluster assignment for embedding $z_i$.

To obtain representative samples, we define the prototype $p_c$ for cluster $c$ as the sample closest to its center:

$$p_c = \arg \min_{i \,:\, c(i)=c} \left\| z_i - \mu_c \right\|_2.$$

These prototypes are retained and fixed throughout the unlearning phase to provide semantic anchors in the embedding space.

For POU, prototype selection is performed in the embedding space of the pretrained ResNet-18 followed by a three-layer residual MLP. These prototypes are used both to define synthetic outlier

targets (based on global min-max bounds) and to verify forgetting via $\sigma$-band outlier detection. In COU, the prototypes are selected from embeddings obtained using a linear projection head trained on frozen DINOv2 or CLIP ViT-L/14 features with supervised contrastive loss. Here, prototypes serve to define task-specific semantics and guide perturbation during the objective removal phase. This shared prototype space provides a consistent geometric reference for both sample-level and objective-level forgetting.

### 3.3 PULL-TO-OUTLIER UNLEARNING (POU)

Pull-to-Outlier Unlearning (POU) removes the influence of specific samples by relocating their embeddings beyond the semantic boundary of the retained dataset. The method updates only the projection head, using a combination of geometric target generation and two complementary loss terms. Below, we describe each stage in detail.

#### 3.3.1 OUTLIER TARGET GENERATION

Let $\{z_i\}_{i=1}^{N} \subset \mathbb{R}^D$ be the projected embeddings of all training samples, and let $\mathcal{U}$ denote the set of indices to be unlearned. We first compute per-dimension bounds over the retained set $\mathcal{N} = \{1, \ldots, N\} \setminus \mathcal{U}$:

$$g_{\min,d} = \min_{j \in \mathcal{N}} z_{j,d}, \quad g_{\max,d} = \max_{j \in \mathcal{N}} z_{j,d}, \quad \text{for } d = 1, \ldots, D.$$

Then, for each $i \in \mathcal{U}$, we define a synthetic outlier target $\tau_i \in \mathbb{R}^D$:

$$\tau_{i,d} = \begin{cases} g_{\min,d} - \delta, & \text{with probability } 0.5, \\ g_{\max,d} + \delta, & \text{otherwise,} \end{cases}$$

where $\delta > 0$ is a fixed margin. This ensures that $\tau_i$ lies outside the retained embedding manifold in all dimensions.

#### 3.3.2 PULL LOSS

We use a pull loss to push each unlearning sample $z_i$ toward its corresponding outlier target $\tau_i$:

$$L_{\text{pull}} = \frac{1}{|\mathcal{U}|} \sum_{i \in \mathcal{U}} \|z_i - \tau_i\|_2^2.$$

This displaces the embeddings of forgotten samples away from high-density regions of the training distribution.

#### 3.3.3 FROZEN LOSS

To maintain stability of the non-target data, we penalize movement of retained embeddings relative to their original positions $z_j^{(0)}$:

$$L_{\text{frozen}} = \frac{1}{|\mathcal{N}|} \sum_{j \in \mathcal{N}} \|z_j - z_j^{(0)}\|_2^2.$$

This regularization preserves the semantic structure of the remaining dataset.

#### 3.3.4 COMBINED OBJECTIVE

The total loss is a weighted combination $L_{\text{POU}} = \lambda_{\text{pull}} \cdot L_{\text{pull}} + \lambda_{\text{frozen}} \cdot L_{\text{frozen}}$. Only the projection head is optimized.

#### 3.3.5 OUTLIER DETECTION CRITERION

To verify successful forgetting, we additionally check whether the final embedding of each unlearned sample lies outside all prototype regions. For each prototype $p_c$, we compute the per-dimension

standard deviation over samples $X_c$ assigned to that cluster as $\sigma_{c,d} = \sqrt{\frac{1}{|X_c|} \sum_{x_i \in X_c} (z_{i,d} - \mu_{c,d})^2}$. We then assess whether $z_i$ deviates from $\mu_c$ in a sufficient number of dimensions:

$$|\{d : |z_{i,d} - \mu_{c,d}| > 3 \cdot \sigma_{c,d}\}| > \tau \cdot D.$$

This 3-sigma criterion serves as a statistical boundary to detect whether a sample has exited the semantic region defined by any cluster. If the condition holds for all $c$, the sample is flagged as successfully forgotten. This mechanism provides an interpretable, prototype-aware certificate of semantic removal.

All retained and test samples are classified using a nearest-prototype classifier, computed by assigning each embedding to the closest prototype with Euclidean distance.

### 3.3.6 EXPERT AGGREGATION STRATEGY

To scale unlearning to large deletion sets while preserving frozen backbone features, POU adopts a Mixture-of-Experts (MoE) design (Jacobs et al., 1991). The forget set is split into disjoint subsets, each handled by a separate residual MLP expert initialized from the pretrained projection head and trained independently using pull and frozen losses.

For retained samples, we apply soft aggregation by averaging outputs from all experts, promoting stability. For unlearned samples, we adopt hard expert routing (Fedus et al., 2021), selecting the expert whose output best matches the sample's outlier target. This design enables scalable, modular unlearning without modifying the backbone.

### 3.4 CONTRASTIVE OBJECTIVE-LEVEL UNLEARNING (COU)

Contrastive Objective-level Unlearning (COU) enables the removal of entire learned objectives or capabilities through projection-only updates. Unlike conventional sample-wise unlearning approaches, COU suppresses a complete representational function by first perturbing the projection space and then restoring it without modifying the backbone network. This process unfolds in two distinct phases.

### 3.4.1 LEARNING WITH ARBITRARY OBJECTIVE

We begin by selecting a subset of training samples $\{x_i\}_{i \in \mathcal{U}}$ and assigning each to a pseudo-target class, chosen as the second-nearest prototype $p_c$ such that $c \neq y_i$. This defines an arbitrary, synthetic objective in the latent space that deviates from the model's original class structure. To impose this new objective, we optimize a perturbation loss that maximizes alignment between the sample and its pseudo-target:

$$L_{\text{perturb}} = \frac{1}{|\mathcal{U}|} \sum_{i \in \mathcal{U}} \left[ 1 - \max_{c \neq y_i} \cos(z_i, p_c) \right],$$

where $z_i$ denotes the projected embedding of $x_i$. This updates the projection head to encode the injected objective, leaving all other parts of the model untouched.

### 3.4.2 UNLEARNING

To remove the influence of the modified objective and revert to the original semantic geometry, we apply a contrastive pull-back mechanism. For each perturbed sample, we restore its embedding to its original location $z_i^{(0)}$ recorded prior to the perturbation phase. The unlearning loss is defined as:

$$L_{\text{restore}} = \frac{1}{|\mathcal{U}|} \sum_{i \in \mathcal{U}} \left[ 1 - \cos(z_i, z_i^{(0)}) \right].$$

Importantly, no additional preservation terms are used for non-target samples, making the method simple, scalable, and focused entirely on forgetting the injected objective.

COU may offer a valuable mechanism for self-improving or agent-driven learning systems, where dynamic objectives are proposed, explored, and revised over time. In such settings, COU may be useful for retracting undesired or unstable objectives introduced by agents, enabling selective unlearning on top of foundational models through lightweight, head-only updates.

Table 1: Instance-level unlearning results on CIFAR10, CIFAR100, and Caltech-256. We randomly subsample 1% of the training set as the unlearning set and use the same set across all baselines. Metrics include: **Rem** (Remaining Set Accuracy ↑), **Unl** (Unlearning Set Accuracy ↓), **Test** (Test Set Accuracy ↑), **TW** (Trustworthiness ↑), and $L_2$ **Drift** (Average $L_2$ Embedding Drift ↓). Arrows indicate whether higher (↑) or lower (↓) values are better.

| Method | CIFAR10 | | | | | CIFAR100 | | | | | Caltech-256 | | | | |
|---|---|---|---|---|---|---|---|---|---|---|---|---|---|---|---|
| | Rem ↑ | Unl ↓ | Test ↑ | TW ↑ | $L_2$ ↓ | Rem ↑ | Unl ↓ | Test ↑ | TW ↑ | $L_2$ ↓ | Rem ↑ | Unl ↓ | Test ↑ | TW ↑ | $L_2$ ↓ |
| Retrain | 99.62% | 86.80% | 88.19% | - | - | 99.24% | 59.00% | 61.98% | - | - | 99.30% | 47.01% | 46.24% | - | - |
| NegGrad | 10.02% | 7.60% | 10.00% | 0.5335 | $> 2 \times 10^8$ | 1.00% | 0.80% | 1.00% | 0.5130 | $> 7.9 \times 10^8$ | 0.65% | 0.00% | 0.57% | 0.5123 | $> 5.5 \times 10^8$ |
| UnrollSGD | 99.44% | 93.08% | 87.03% | 0.9900 | 1.3633 | 98.78% | 99.80% | 60.91% | 0.9931 | 1.3466 | 99.42% | 100.00% | 43.45% | 0.9884 | 6.7948 |
| SISA | 99.52% | 100.00% | 88.01% | 0.9941 | 7.2353 | 99.30% | 96.90% | 61.70% | 0.9992 | 6.0514 | 86.86% | 80.59% | 45.28% | 0.9816 | 7.6901 |
| LAF | 86.43% | 83.20% | 74.62% | 0.4995 | 3.5835 | 57.31% | 54.40% | 34.34% | 0.5008 | 6.1496 | 60.73% | 36.07% | 46.40% | 0.4989 | 3.6107 |
| **POU (Ours)** | 99.07% | 0.00% | 86.89% | 1.0000 | 0.9451 | 98.58% | 0.00% | 61.41% | 1.0000 | 1.3388 | 98.25% | 0.00% | 39.02% | 1.0000 | 0.8585 |

Table 2: Class-level unlearning results on CIFAR100 and Caltech-256 using POU and baseline methods. We consistently select the 5th class for unlearning across all experiments.

| Method | CIFAR100 | | | | | Caltech-256 | | | | |
|---|---|---|---|---|---|---|---|---|---|---|
| | Rem ↑ | Unl ↓ | Test ↑ | TW ↑ | $L_2$ ↓ | Rem ↑ | Unl ↓ | Test ↑ | TW ↑ | $L_2$ ↓ |
| Retrain | 99.03% | 0.00% | 61.39% | - | - | 98.43% | 0.00% | 44.42% | - | - |
| NegGrad | 1.00% | 0.00% | 1.00% | 0.5349 | $> 8.2 \times 10^4$ | 0.57% | 0.00% | 0.65% | 0.5117 | $> 5.2 \times 10^8$ |
| UnrollSGD | 99.24% | 97.26% | 50.10% | 0.9887 | 6.7679 | 98.52% | 96.33% | 44.83% | 0.9884 | 6.7584 |
| LAF | 80.45% | 100.00% | 46.35% | 0.5022 | 2.1888 | 98.19% | 98.89% | 45.03% | 0.4976 | 4.3002 |
| **POU (Ours)** | 98.58% | 0.00% | 60.99% | 1.0000 | 0.9662 | 98.27% | 0.00% | 39.02% | 1.000 | 0.9609 |

# 4 EXPERIMENTS

## 4.1 DATASETS

We evaluate our proposed unlearning methods on three standard image classification benchmarks. For **CIFAR10** (Krizhevsky, 2009), POU is applied to the full dataset of 60,000 images across 10 classes, while COU uses a subsample of 10,000 images (1,000 per class) for fine-grained control over learned objectives. For **CIFAR100** (Krizhevsky, 2009), we use the complete set of 60,000 images across 100 fine-grained categories to test POU under greater class granularity and clustering density. **Caltech-256** (Griffin et al., 2007) includes 30,607 images from 256 diverse categories and is used exclusively for POU to assess unlearning in high-variability domains with complex prototype and semantic structures. For all datasets, we split 90% of the data for training and 10% for testing.

## 4.2 EVALUATION METRICS

To evaluate both the effectiveness of unlearning and the preservation of retained knowledge, we adopt distinct metrics for instance-level unlearning (POU) and objective-level unlearning (COU). For POU, we report: (1) **Remaining Set Accuracy**, the classification accuracy on retained samples, where higher scores indicate preserved knowledge; (2) **Unlearning Set Accuracy**, the accuracy on deleted samples, where values near 0% reflect effective forgetting; (3) **Test Set Accuracy**, measuring performance on the original test set after unlearning; (4) **Trustworthiness** (Venna & Kaski, 2001), which evaluates local neighborhood consistency between the original and final embeddings of retained data; and (5) $L_2$ **Drift**, the average Euclidean distance between the original and updated retained embeddings, quantifying collateral change.

For COU, we compute: (1) **Trustworthiness (Post-Perturbation & Post-Unlearning)**, calculated over all samples to assess the projection space's structural changes and recovery; and (2) $L_2$ **Drift (Post-Perturbation & Post-Unlearning)**, representing embedding displacement induced by perturbation and reduced through unlearning.

## 4.3 BASELINES

**Retrain:** The most intuitive unlearning baseline involves full model retraining on the retained dataset after the removal of target samples. While it provides an exact removal guarantee, it is computationally infeasible in practice and scales poorly to frequent deletions.

Table 3: Objective-level unlearning results on a subset of CIFAR10 using COU and NegGrad, evaluated on frozen DINOv2 and CLIP ViT-L/14 features. Metrics: Trustworthiness (TW) after perturbation and unlearning phases, and $L_2$ Drift after unlearning.

| Method | DINOv2 ViT-L/14 | | | CLIP ViT-L/14 | | |
|---|---|---|---|---|---|---|
| | TW Pert | TW Unlearn $\uparrow$ | $L_2 \downarrow$ | TW Pert | TW Unlearn $\uparrow$ | $L_2 \downarrow$ |
| NegGrad | 0.9747 | 0.9699 | 201.3660 | 0.9845 | 0.9607 | 93.3761 |
| **COU (Ours)** | 0.9747 | 1.0000 | 1.6496 | 0.9845 | 0.9999 | 0.8286 |

**NegGrad:** This method approximates forgetting by fine-tuning the model with reversed gradients on the forget set. For COU, we report results using NegGrad as the sole baseline, although it was not originally designed for objective-level unlearning and lacks explicit mechanisms for forgetting latent objectives or projection-specific behavior.

**Unrolling SGD** (Thudi et al., 2021): UnrollSGD leverages optimization trajectory inversion by tracking and reversing gradient steps associated with the target samples. It achieves approximate unlearning with bounded error but requires storing past gradients and optimizer states, making it memory-intensive.

**SISA** (Bourtoule et al., 2019): Sharded, Isolated, Sliced, and Aggregated (SISA) training splits the dataset into multiple shards and slices to localize retraining when unlearning is required.

**Label-Agnostic Forgetting (LAF)** (Shen et al., 2024): A recent unsupervised method that removes entire classes via variational distribution matching without relying on explicit labels. LAF demonstrates that principled removal is possible even without supervision.

### 4.4 IMPLEMENTATION DETAILS

All experiments were conducted on a machine equipped with an AMD Ryzen 9 5900HX CPU, 32 GB RAM, and an NVIDIA RTX 3080 Laptop GPU with 16 GB memory.

We use a batch size of 64 and a learning rate of $1e-4$ across all experiments. All baseline methods are trained for 100 epochs, while the ResNet-18 + MLP backbone used in POU and Retrain is pretrained for 120 epochs. For **POU**, we use the Adam optimizer and set $\lambda_{\text{pull}} = 1.0$, $\lambda_{\text{frozen}} = 50.0$, and margin $\delta = 20.0$. We apply early stopping when both the pull loss value drops below 10 and the frozen loss falls below 0.02, indicating sufficient displacement of unlearned samples and stability of retained embeddings. For **COU**, we also use Adam and train the linear projection head for 5 epochs using a supervised contrastive loss with temperature 0.07, and stop unlearning once the mean $L_2$ drift of pulled-back samples falls below 2.

For the baselines: **SISA** uses 5 shards, 2 slices per shard, and 20 epochs per slice. **LAF** is trained with 20 VAE epochs and 5 unlearning epochs. **NegGrad** uses $\lambda_{\text{neg}} = 1.0$ for gradient ascent on the forget set. **UnrollSGD** performs 5 epochs of SGD trajectory recording and applies influence reversal over 50 unroll steps.

### 4.5 INSTANCE AND CLASS-LEVEL UNLEARNING RESULTS

We evaluate instance-level and class-level forgetting using POU across CIFAR10, CIFAR100, and Caltech-256. As shown in Table 1 and Table 2, POU consistently achieves perfect forgetting on the unlearned subset (0% accuracy), while preserving accuracy on the remaining set and test set. In contrast, prior baselines struggle to eliminate the target information without causing significant collateral damage to the rest of the model. Notably, the slight accuracy drop of POU on the remaining and test sets compared to Retrain may stem from the difference in classification strategy, as POU uses a nearest-prototype classifier instead of a standard softmax classification head; however, its perfect trustworthiness score indicates that the underlying representation structure is fully preserved.

Trustworthiness and $L_2$ drift metrics further reveal that POU maintains the integrity of the retained representation space. These claims are visually supported in Figure 3, where POU preserves compact cluster structure while cleanly displacing unlearned samples to synthetic outlier regions. Sim-

ilarly, Figure 5 confirms that even when entire classes are forgotten, the semantic organization of remaining data remains intact.

We also evaluated robustness using a Membership Inference Attack (MIA) (Table 4). For Attack A (remain vs. test), methods like UnrollSGD reduce leakage closer to chance, while POU maintains stable behavior for retained data, ensuring representation preservation. For Attack B (forget vs. test), POU is clearly the best, driving leakage down to near-chance (2.89%) and thus providing the strongest guarantee of effective forgetting. This highlights that POU achieves the best balance between utility and privacy.

### 4.6 OBJECTIVE-LEVEL UNLEARNING RESULTS

In objective-level unlearning, the goal is to remove an injected latent capability rather than specific samples. Table 3 demonstrates that COU effectively erases such injected objectives, restoring the projection space to its original structure. This is achieved without modifying the frozen backbone, relying solely on lightweight projection-level updates.

NegGrad, by contrast, shows an inability to reverse the impact of the learned objective, resulting in high drift and a loss of semantic separability. Figure 4 visualizes this difference: COU successfully restores original class-wise clusters after unlearning, while NegGrad collapses them into entangled or flattened manifolds. These results indicate that COU provides the first effective solution for capability-level forgetting via projection-only tuning.

### 4.7 DISCUSSION

#### 4.7.1 ABLATIONS

To understand the role of each component in POU, we conduct an ablation study focused on the frozen loss, as reported in Table 5. When the frozen loss is removed, forgetting becomes incomplete and unlearned samples remain entangled with retained data. Although performance on the retained set may appear high, the underlying structure is destabilized, leading to high drift and reduced trustworthiness.

In contrast, incorporating the frozen loss consistently maintains the geometric alignment of retained embeddings while allowing targeted forgetting of unwanted samples. This term proves essential for balancing deletion and preservation objectives within the same embedding space.

#### 4.7.2 VISUAL ANALYSIS OF STRUCTURAL PRESERVATION

Figures 3, 4, and 5 provide qualitative evidence supporting the numerical results. In instance-level unlearning, POU visibly displaces unlearned samples far from their original clusters without disrupting the surrounding distribution. This demonstrates that forgetting is both complete and localized.

For objective-level unlearning, COU recovers the original projection layout after removing the arbitrary objective, confirming that projection-space perturbations are reversible under our design. In contrast, NegGrad leads to significant semantic collapse and deformation. These visualizations underscore the importance of geometric regularity and prototype alignment in designing effective unlearning mechanisms.

## 5 CONCLUSION

We introduced POCO, a unified framework for sample-level and objective-level machine unlearning through projection-only techniques. Our POU method enables efficient instance and class forgetting by geometrically displacing unwanted embeddings, while COU removes entire objectives by perturbing the projection space and then recovering its original structure. Both methods operate above frozen feature extractors, offering scalable and interpretable forgetting without retraining. Extensive evaluations on CIFAR10, CIFAR100, and Caltech-256 confirm that POCO achieves high retention, minimal drift, and reliable forgetting. These results demonstrate the potential of our approach for enabling efficient, interpretable unlearning in privacy-preserving and adaptable machine learning systems.

## REPRODUCIBILITY STATEMENT

We ensure reproducibility by providing implementation details in Section 4.4, including training schedules, loss functions, hyperparameters, and early stopping criteria. All datasets (CIFAR10, CIFAR100, and Caltech-256) are publicly available, with preprocessing steps included in the supplementary code. Pseudocode for POU and COU is given in Algorithms 1 and 2, and an anonymized implementation is included in the supplementary material for direct reproduction.

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

# A APPENDIX

Table 4: Threshold-based Membership Inference Attack (MIA) AUC (%): Attack A = remain vs. test; Attack B = forget vs. test. Lower is better.

| Method | AUC-A Before | AUC-A After | AUC-B Before | AUC-B After |
|--------|--------------|-------------|--------------|-------------|
| POU | 59.34% | 59.75% | 60.06% | **2.89%** |
| NegGrad | 61.51% | 50.26% | 61.69% | 49.05% |
| unroll_sgd | 63.01% | 49.68% | 63.46% | 49.96% |
| LAF | 61.79% | 58.82% | 63.95% | 62.72% |
| SISA | 62.94% | 62.91% | 61.98% | 61.84% |

Table 5: Ablation study of the POU method. We compare the full method with the variant that removes the $L_{\text{frozen}}$ term across CIFAR10, CIFAR100, and Caltech-256. Metrics: Rem ($\uparrow$), Unl ($\downarrow$), Test ($\uparrow$), TW ($\uparrow$), and $L_2$ Drift ($\downarrow$).

**(a) CIFAR10**

| Setting | Rem $\uparrow$ | Unl $\downarrow$ | Test $\uparrow$ | TW $\uparrow$ | $L_2 \downarrow$ |
|---------|------|------|------|------|------|
| POU without $L_{\text{frozen}}$ | 99.23% | 22.45% | 86.31% | 0.9971 | 168.78 |
| POU with $L_{\text{frozen}}$ | 99.07% | 0.00% | 86.89% | 1.0000 | 0.95 |

**(b) CIFAR100**

| Setting | Rem $\uparrow$ | Unl $\downarrow$ | Test $\uparrow$ | TW $\uparrow$ | $L_2 \downarrow$ |
|---------|------|------|------|------|------|
| POU without $L_{\text{frozen}}$ | 93.95% | 16.34% | 58.22% | 0.9972 | 197.18 |
| POU with $L_{\text{frozen}}$ | 98.58% | 0.00% | 61.41% | 1.0000 | 1.34 |

**(c) Caltech-256**

| Setting | Rem $\uparrow$ | Unl $\downarrow$ | Test $\uparrow$ | TW $\uparrow$ | $L_2 \downarrow$ |
|---------|------|------|------|------|------|
| POU without $L_{\text{frozen}}$ | 75.47% | 7.82% | 26.23% | 0.9820 | 329.42 |
| POU with $L_{\text{frozen}}$ | 98.25% | 0.00% | 39.02% | 1.0000 | 0.86 |

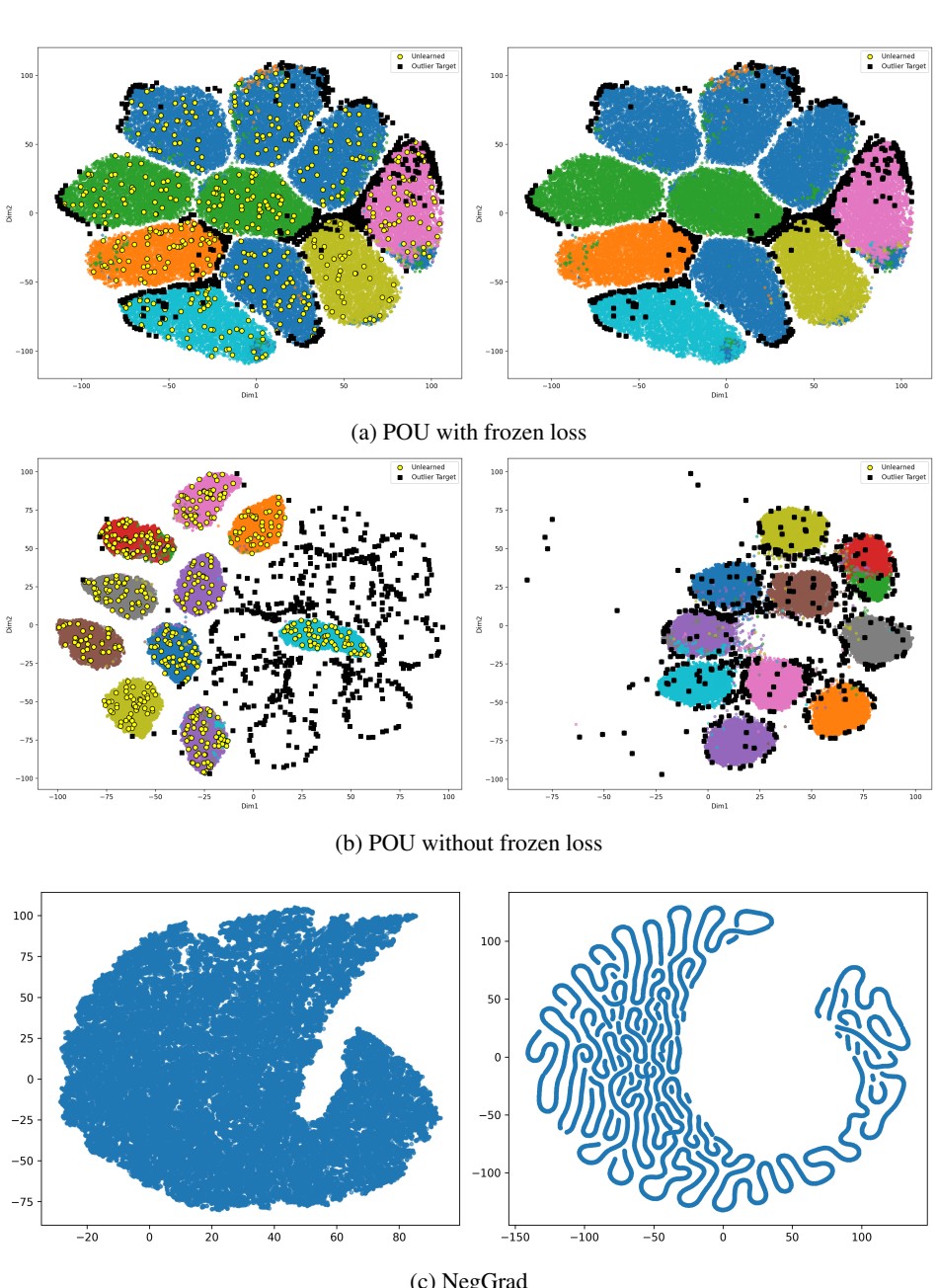

Figure 3: t-SNE visualizations of instance-level unlearning on CIFAR10. The comparison illustrates the importance of the frozen loss in POU. With frozen loss (top), cluster structures are preserved and unlearned samples are cleanly displaced to outlier regions. Without frozen loss (middle), forgetting is incomplete and retained clusters are distorted. NegGrad (bottom) shows severe representation collapse and poor separation.

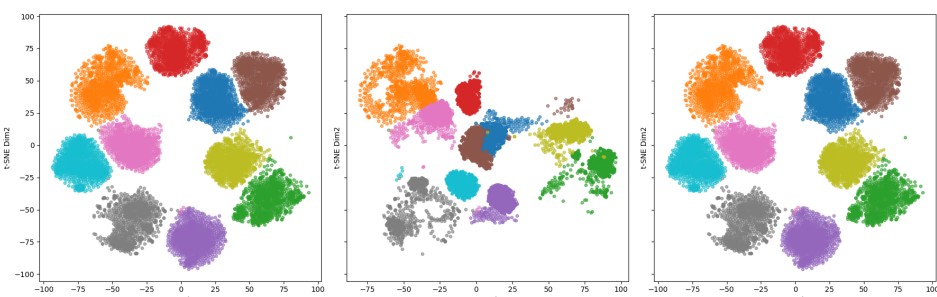

(a) COU: t-SNE visualizations before and after objective-level unlearning on DINOv2 features. Left: original embedding space. Middle: after injecting arbitrary objective. Right: after COU unlearning. COU restores clean cluster structure.

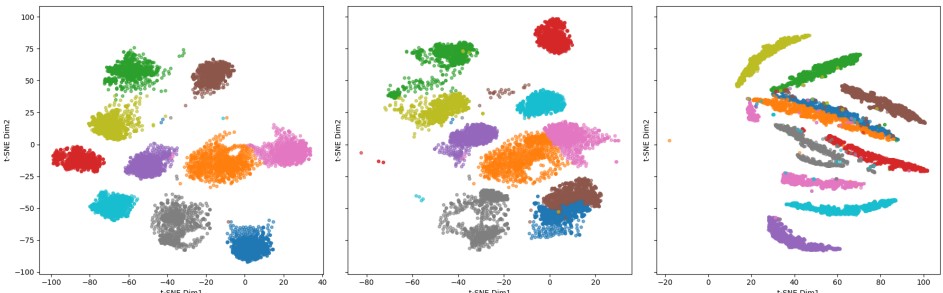

(b) NegGrad: t-SNE visualizations before and after objective-level unlearning on DINOv2 features. Left: original embedding space. Middle: after injecting arbitrary objective. Right: after NegGrad-based unlearning. Class structure is not recovered.

Figure 4: Qualitative comparison of COU and NegGrad for objective-level unlearning on DINOv2 embeddings. Only COU restores the original semantic structure of the projection space after removing injected objectives.

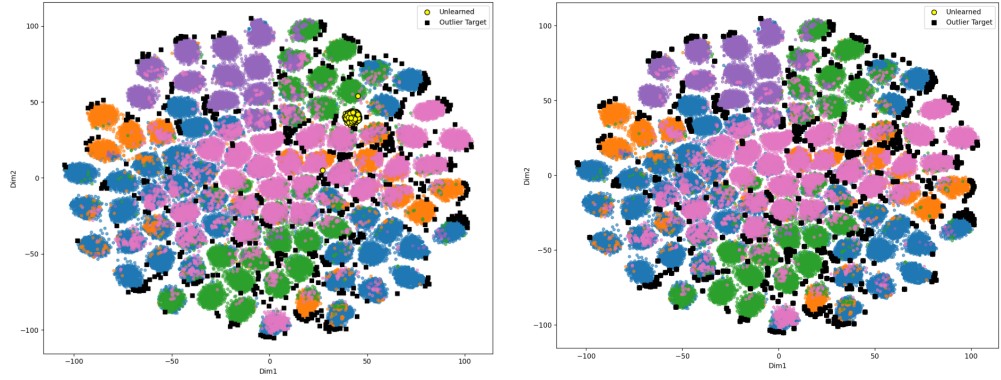

Figure 5: t-SNE visualization of class-level unlearning on CIFAR100 using POU. **Left:** The original projection space before unlearning shows well-separated latent subclusters for each class. **Right:** After unlearning, the deleted class samples (yellow) are successfully displaced to synthetic outlier regions (black squares), while the rest of the latent structure remains stable. This confirms that POU achieves class forgetting without disturbing non-target data.

---

**Algorithm 1** Pseudocode for Pull-to-Outlier Unlearning (POU)

---

**Input:** Pretrained MLP head $f_\theta$, training features $\mathbf{Z}$, forget indices $\mathcal{U}$, retained indices $\mathcal{N}$, margin $\delta$, number of experts $K$
**Initialize:** Copy $K$ expert MLPs from $f_\theta$ as $\{\theta_k\}_{k=1}^{K}$
Split $\mathcal{U}$ into $K$ disjoint chunks: $\mathcal{U}_1, \ldots, \mathcal{U}_K$
**for** $k = 1$ **to** $K$ **do**
    Compute global bounds $g_{\min}, g_{\max}$ from retained set $\mathcal{N}$
    For each $i \in \mathcal{U}_k$, compute outlier target $\tau_i$ using $\delta$, $g_{\min}$, $g_{\max}$
    Optimize $\theta_k$ on:
        Pull loss $\mathcal{L}_{\text{pull}} = \sum_{i \in \mathcal{U}_k} \|f_{\theta_k}(z_i) - \tau_i\|^2$
        Frozen loss $\mathcal{L}_{\text{frozen}} = \sum_{j \in \mathcal{N}} \|f_{\theta_k}(z_j) - f_\theta(z_j)\|^2$
        Total loss: $\mathcal{L}_{\text{POU}} = \lambda_{\text{pull}} \cdot \mathcal{L}_{\text{pull}} + \lambda_{\text{frozen}} \cdot \mathcal{L}_{\text{frozen}}$
**end for**
**Aggregation:**
    For retained samples, average outputs from all experts
    For unlearned samples, select best expert based on closest match to $\tau_i$
**Evaluation:** Run nearest-prototype classification and $3\sigma$ outlier check

---

**Algorithm 2** Pseudocode for Contrastive Objective-Level Unlearning (COU)

---

**Input:** Pretrained projection $f_\theta$, features $\mathbf{X}$, labels $\mathbf{y}$, forget set $\mathcal{U}$, prototypes $\{p_c\}$, epochs $E$
**Stage 1: Learning Arbitrary Objective**
**for** $e = 1$ **to** $E$ **do**
    For each $i \in \mathcal{U}$:
        Compute second-nearest prototype $p_{c'}$, $c' \neq y_i$
        Apply perturbation loss:

$$\mathcal{L}_{\text{perturb}} = \sum_{i \in \mathcal{U}} \left[ 1 - \max_{c \neq y_i} \cos(f_\theta(x_i), p_c) \right]$$

    Update $\theta$ via $\mathcal{L}_{\text{perturb}}$
**end for**
**Stage 2: Unlearning**
For each $i \in \mathcal{U}$:
    Retrieve original embedding $z_i^{(0)}$
    Apply contrastive pull-back loss:

$$\mathcal{L}_{\text{restore}} = \sum_{i \in \mathcal{U}} \left[ 1 - \cos(f_\theta(x_i), z_i^{(0)}) \right]$$

Update $\theta$ via $\mathcal{L}_{\text{restore}}$
**Evaluation:** Measure trustworthiness and $L_2$ drift on full dataset before and after both stages

---

