# OpenReview forum: "A Framework for Sample and Objective Forgetting: Pull-to-Outlier & Contrastive Objective-level (POCO) Unlearning"
_ICLR.cc/2026/Conference — ICLR 2026 Conference Withdrawn Submission_

### Official Review · Reviewer_YQm9 · 2025-10-26

**Soundness:** 1
**Presentation:** 2
**Contribution:** 2
**Rating:** 2
**Confidence:** 3

**Summary:**

This paper introduces two frameworks for machine unlearning Prototype-based Outlier Unlearning (POU) and Contrastive Objective-level Unlearning (COU) to unlearn samples and an entire class, respectively.

POU removes target samples by pushing their embeddings outside the known data manifold and it employs a mixture-of-experts design that maps samples to unlearn to synthetic outliers beyond the retained data embedding bounds.

COU removes target class by breaking class associations to corresponding prototypes and then restoring the associations for retained data ensuring class separability for all remaining non-target classes.

The authors conducted experiments on CIFAR10, CIFAR100, CalTech256 using ResNet-18 (for POU) and DINOv2 and CLIP ViT-L/14 (for COU) backbones and compared methods using Remaining Set Accuracy, Unlearning Set Accuracy, Test Set Accuracy, Trustworthiness (Venna & Kaski,
2001), which evaluates local neighborhood consistency between the original and final embeddings
of retained data and L2 Drift.

**Strengths:**

- The paper tackles an important problem and the motivation is well justified
- Codes and hyper-parameter values are provided
- The POU section is well-written and clear to understand

**Weaknesses:**

1) Ablations are incomplete:
- Removing Lpull i.e.,
- Studying the impact of different values for POU hyper-parameters (loss weights, sigma for outliers, and margin)
- Sensitivity to deeper backbones other e.g., ResNet-50 or -101
- Number of experts, right now it is not clear how many were used for the current results
- Removing stage 1 from COU
- Removing stage 2 from COU
- Sensitivity to number of classes to be reomved using COU


2) Some of the results are missing for COU including: remaining set accuracy, unlearning accuracy, test set accuracy, and. MIA.

3) I am not convinced that COU should be compared only with NegGrad. Even if a method is not explicitly designed for objective-level unlearning still it is necessary to show that the idea is competitive compared to other approaches to class unlearning.

I'd be happy to raise my rating if authors can address these weaknesses and my questions regarding how these methods work particularly COU.

**Questions:**

- How the forget set was splitted into disjoint subsets for POU?

- In POU, why do you need both class centers and class prototypes? why can’t you only use the centers? it seems to me that prototypes are irrelevant to the method because the only usage is at the end for classification which can also be done by centers

- Help me understand COU better, first we perturb a subset of the forget set (denoted by U) and then we try to retrieve broken associations similar to a denoising autoencoder. How does this help with unlearning a class or a behaviour as claimed in the paper? Section 3.4 defines U as a training set but the rest of the paper and algorithm 2 defines it as the forget set. It makes more sense that we break all associations and then retrieve the non-target ones but right now it is not clear based on the paper if that’s what’s happening.

- It seems POU can be a baseline for COU because we can remove all samples associated with the target class. What prevents us from applying POU in a setup like that and compare it to COU?  similarly, why can’t we apply COU only to a subset of samples in a class?

- What happens if we remove the majority of samples from a class using POU? Does the per class performance remain competitive?


- Authors mentioned that 90% of the data was for train and 10% for test, can you elaborate on how did you optimize for hyper-parameters without a validation set?

- How many times did you run each experiment? please report standard error or confidence interval as some of the numbers are very close

---

### Official Review · Reviewer_ooxf · 2025-10-28

**Soundness:** 2
**Presentation:** 2
**Contribution:** 1
**Rating:** 2
**Confidence:** 4

**Summary:**

The paper proposes a projection-only framework operating above frozen feature extractors, targets two levels of machine unlearning (MU). First, Pull-to-Outlier Unlearning (POU) for instance/class forgetting by pushing forget-set embeddings beyond the in-distribution manifold with a projection head and a frozen-loss to preserve retained data. Second, Contrastive Objective-level Unlearning (COU) to remove an entire learned objective/capability by perturbing and then restoring a contrastive projection head trained on frozen DINO/CLIP features. The method includes a prototype-based outlier criterion as a transparent verification of forgetting and scales via a mixture-of-experts head. Experiments on CIFAR-10/100 and Caltech-256 claim near-perfect forgetting with limited drift for POU, and improved trustworthiness/L2-drift for COU.

**Strengths:**

1. The combination of pull and frozen in POU is intuitive, and the COU pipeline is concise. Both are compatible with frozen feature extractors, and the paper provides implementation details and pseudocode.

2. The prototype-based $3\sigma$ rule offers a transparent geometric, per-example criterion that strengthens unlearning auditing.

3. Experiments and visualizations support the effectiveness of both POU and COU.

**Weaknesses:**

1. The setting of freezing the feature extractor, pushing samples beyond prototype ranges, and performing label-free unlearning has prior art. Work [a] achieves class-level unlearning by directly applying a linear filtration matrix in logit space. Boundary Unlearning [b] pushes forget samples across the decision boundary. This makes the contribution of this paper comparatively minor. In addition, several important baselines (e.g., [c, d]) are missing from the experiments. The paper should broaden its literature coverage on unlearning and clarify how the proposed methods differ from prior work.

2. The paper mentions sample-level, class-level, and the objective-level unlearning that it focuses on, but does not formally define how objective-level differs from sample/class-level, nor does it provide relevant citations. This leaves readers confused and risks misunderstanding what “objective-level unlearning” entails.

3. In §3.4 (COU), two stages are outlined, “Learning with Arbitrary Objective” and then “Unlearning”, but the role of the first stage is unclear. The pseudo-target does not appear to be a standard training objective. The paper should clearly articulate COU’s intended application scenarios and the motivation for this two-stage design.

4. The proposed $3\sigma$ criterion could be exploited by an adversary to aid membership inference attacks (MIA) by identifying the presence of forgetting samples. Although MIA experiments are reported, the paper does not describe attack implementation details or analyze whether such threats from an intended attacker could undermine the proposed approach.

5. Freezing the feature extractor and retraining a prototype classifier restricts applicability only to classification, limiting extensibility to broader domains such as image generation or large language models.

### Reference

a. Baumhauer, Thomas et al. “Machine unlearning: linear filtration for logit-based classifiers.” Machine Learning (2020).

b. Chen, Min et al. “Boundary Unlearning: Rapid Forgetting of Deep Networks via Shifting the Decision Boundary.” CVPR (2023).

c. Fan, Chongyu et al. “SalUn: Empowering Machine Unlearning via Gradient-based Weight Saliency in Both Image Classification and Generation.” ICLR (2024).

d. Foster, Jack et al. “Fast Machine Unlearning Without Retraining Through Selective Synaptic Dampening.” ArXiv (2023).

**Questions:**

1. The paper does not specify how “the forget set is split into disjoint subsets” (line 284), nor does it include an ablation to demonstrate the effect of using MoE.

2. The sensitivity study is insufficient. There is no ablation on the number of clusters/prototypes or on the margin hyper-parameter $\sigma$.

---

### Official Review · Reviewer_aq9s · 2025-10-31

**Soundness:** 2
**Presentation:** 2
**Contribution:** 3
**Rating:** 2
**Confidence:** 4

**Summary:**

This paper proposes a unified framework for machine unlearning built upon frozen backbone features. The framework contains two complementary modules: Pull-to-Outlier Unlearning (POU) and Contrastive Objective-level Unlearning (COU). The paper evaluates POU and COU on CIFAR10, CIFAR100, and Caltech-256 using frozen DINOv2/CLIP features.

**Strengths:**

(1)Both POU and COU avoid expensive backbone retraining.

(2)POU’s idea of pushing forgotten samples “outside the manifold” is simple and intuitive.

(3)The perturbation of task semantics followed by a geometric restoration is a conceptual structure for unlearning at the objective level.

**Weaknesses:**

(1)Although POU avoids backbone retraining, it requires running K-means clustering on the entire retained dataset to construct the prototype space. For large-scale applications with millions of embeddings, this step becomes the computational bottleneck and may outweigh the benefit of avoiding retraining. The paper does not discuss the scalability implications or provide strategies such as streaming clustering, approximate K-means, or prototype distillation to address this limitation.

(2)The frozen-loss term demands storing and revisiting all original embeddings z_j^(0)​, leading to substantial memory and computation overhead.

(3)The MoE-based expert training and prototype-based verification scale linearly with the number of clusters and the size of the forget set, further increasing the cost.

(4)In POU, the outlier target design is arbitrary and lacks theoretical justification for guaranteeing complete forgetting.

(5)A practical limitation is that POU requires a trainable projection head. In many real-world settings only a frozen backbone is available, meaning that users must construct and train an additional head before unlearning, which weakens the claimed lightweight nature of the approach.

(6)The 3-sigma outlier detection criterion relies on overly strong Gaussian and per-dimension independence assumptions, which do not hold in high-dimensional embedding spaces, making the forgetting certification heuristic and potentially unreliable.

(7)The COU mechanism does not actually remove an existing model capability; instead, it injects and then cancels out a synthetic perturbation at the projection head, making the claimed ‘objective-level unlearning’ logically unsupported.

(8)Both POU and COU explicitly claim support for class-level forgetting, yet the paper does not present any experiments that evaluate or compare their behavior on this setting.

(9)COU does not provide any per-example verification mechanism at all—there is no metric for assessing whether a capability has actually been removed.

(10)While POU is indeed label-free in its mechanism, COU fundamentally depends on a projection head trained with supervised contrastive learning, making it incompatible with fully unsupervised settings. Moreover, both modules still require users to explicitly specify which samples or classes should be forgotten.

(11)POU achieves perfect forgetting (0.00% unlearning accuracy) on all datasets while simultaneously preserving almost the full accuracy of the retained set and maintaining TW=1.000 across all experiments. Such a combination—complete forgetting with no measurable side effects—is inconsistent with well-established trade-offs in the unlearning literature and is not achievable by existing methods. The unlearning is better than the golden standard “Retrain”.

(12)Table 3 evaluates COU only against NegGrad, which is fundamentally a sample-level unlearning method and not designed for objective-level forgetting. As a result, the comparison is not meaningful for the task being evaluated. Moreover, several relevant baselines discussed in the related work section, such as UnrollSGD, SISA, and especially Label-Agnostic Forgetting (LAF), which explicitly targets class-level or capability-level removal, are entirely missing from the experiments. This omission significantly undermines the empirical validation of COU.

(13)The ablation is narrowly focused on removing the frozen loss and lacks breadth (no hyper-parameter sweeps, no alternative regularizers, no runtime analysis, and no class/seed/ratio variants), so it does not convincingly establish the necessity of the proposed component nor rule out simpler explanations.

**Questions:**

(1)In the abstract, the claim that current Machine Unlearning (MU) methods ‘focus only on sample-level forgetting’ is incomplete. As noted in the Introduction, existing methods address both sample-level and class-level forgetting.

(2)The equations in the paper are not numbered, which affects readability and clarity.

(3)The term z_j^(0)​ is introduced without a clear explanation of its meaning.

(4)Figure 2 is introduced but never properly referenced or explained in the main text, leaving its role unclear and weakening the conceptual understanding of COU.

---

### Official Review · Reviewer_cNjN · 2025-10-31

**Soundness:** 2
**Presentation:** 1
**Contribution:** 2
**Rating:** 2
**Confidence:** 3

**Summary:**

This paper considers the problem of machine unlearning. This paper proposes two techniques for unlearning: pull-to-outlier unlearning (POU) and contrastive objective-level unlearning (COU). Between them, COU is claimed to be the first method to achieve objective-level unlearning. Experimental results show the effectiveness of the proposed methods on image datasets.

**Strengths:**

+ Machine unlearning is a timely topic, and objective-level unlearning is interesting.

+ The proposed methods seem to be lightweight.

**Weaknesses:**

- From L220, it is unclear how POU and COU share prototypes: they operate in different embedding spaces.

- While this paper tackles the limitation of sample-level or class-level removal (limitation 1), one of the proposed method, POU still shares the limitation. The other one, COU address this, but only a portion of this paper describes this, making the contribution of this work somewhat weak and deviated.

- While this paper states the lack of fully unsupervised forgetting (limitation 4), the proposed method, COU is supervised, in that the prototype is chosen to be c \neq y_i, i.e., y_i is considered when computing L_perturb.

- Some notations are not defined or overriden; for example, \tau in L273 is different from synthetic outlier target with a subscript while never defined, and c is used as a cluster label in Section 3.2, but later used as class label in Section 3.4.

- How is the forget set organized?

- While there are many hyperparameters, some of them are undefined, making this work irreproducible; i.e., Section 4.4 is insufficient to ensure reproducibility. These include (not exhaustively) the number of clusters C for K-means or the number of experts in POU. In particular, it is likely that the performance of POU might depend on the choice of C, no hyperparameter tuning experiment is provided.

- Some design choices are not justified. For example, why are L2 and cosine losses chosen in Section 3.3 and 3.4, respectively?

- No computational cost comparison to justify the claim that updating only the projector is efficient.

- The idea of training projection only has already been studied in previous works, e.g., [Yu et al.].

[Yu et al.] LegoNet: A Fast and Exact Unlearning Architecture. arXiv:2210.16023

- The effectiveness of the proposed objective-level unlearning is validated under L_perturb only. The authors may want to consider other standard learning objectives to show the generalizability of the proposed COU.

- Figure 4a and 4c seem to be exactly the same; there might be a mistake.

- Experimental results are mostly in the appendix. In principle, the main paper with 9 pages should be standalone, but this paper appears to be not.

**Questions:**

Please address concerns in Weaknesses above.

---

### Note · Authors · 2025-11-12

I have read and agree with the venue's withdrawal policy on behalf of myself and my co-authors.